# Volume Deformation and Hydration Behavior of Ordinary Portland Cement/Calcium Sulfoaluminate Cement Blends

**DOI:** 10.3390/ma16072652

**Published:** 2023-03-27

**Authors:** Guangxiang Ji, Hafiz Asad Ali, Keke Sun, Dongxing Xuan, Xiaoqin Peng, Jingjun Li

**Affiliations:** 1Department of Hydraulic Engineering, Tsinghua University, Beijing 100084, China; 2Department of Civil and Environmental Engineering, The Hong Kong Polytechnic University, Hong Kong 999077, China; 3School of Civil Engineering, Inner Mongolia University of Science and Technology, Baotou 014010, China; 4College of Materials Science and Engineering, Chongqing University, Chongqing 400044, China

**Keywords:** ordinary Portland cement, calcium sulfoaluminate cement, volume deformation, hydration, microstructure

## Abstract

Blends of ordinary Portland cement (OPC) and calcium sulfoaluminate (CSA) cement can be used to adjust the properties of cement for specific applications. In this study, CSA cement was used as a shrinkage-compensating admixture to improve the hydration behavior and performance (compressive strength and drying shrinkage) of OPC; the expansion behavior of the blended cement mortar was evaluate based on the saturation index of ettringite. The experimental results showed that incorporating CSA cement resulted in a delayed effect on the hydration of C_3_S, shortened the induction periods of the blended cement and decreased the setting time. The CSA cement also improved the early compressive strength and drying shrinkage of the OPC due to its compact microstructure. The drying shrinkage of the OPC mortar decreased by 27.8% when 6% CSA cement was used, but the formation of microcracks due to expansion could negatively impact its late compressive strength development and associated pore structures of the blends when the replacement content of CSA cement exceeded 6 wt.%. The results relevant to the expansion behavior of the CSA cements could induce crystallization stress, enhancing its resistance against shrinkage cracking.

## 1. Introduction

Reducing energy consumption and greenhouse gas emissions in the process of clinker production has become an urgent problem that the ordinary Portland cement (OPC) industry needs to address. Some cement-less or low-alkali cements have been produced to reduce the clinker content [1,2,3]. Calcium sulfoaluminate (CSA) cement is considered a low-energy cement that requires a low calcination temperature (1200 °C) and limestone content during manufacturing, resulting in low carbon, sulfur dioxide and oxynitride footprints [4,5]. The CSA clinker is mainly composed of ye’elimite (C_4_A_3_$\bar{S}$, where C = CaO, A = Al_2_O_3_, S^−^ = SO_3_), varying proportions of alite (C3S) or belite (C2S) and calcium aluminate [6]. The hardening processes can occur ambiguously and differ significantly when using a CSA cement clinker or an alite-containing cement clinker, with the mineralogical composition and basicity being two important indicators of the hydration and durability of the clinker [7,8]. Due to C_4_A_3_$\bar{S}$ being very highly reactive, CSA cement has many advantages, including rapid setting, high early strength, glass fiber compatibility and sulfate corrosion resistance [9,10]. However, due to the high cost of procuring high-quality alumina sources, the large-scale application of CSA cement is still restricted [11,12,13].

Due to the intrinsic nature of CSA clinkers, they could potentially be utilized as admixtures for OPC to improve the performance of the resulting binders and result in relatively lower environmental impacts. It was explored that CSA cement can significantly improve the setting time and compressive strength of OPC [14,15], and that OPC/CSA cement blends could be used to enhance the durability and strength development at cold temperatures and in seawater and sewer environments [16,17,18,19]. Furthermore, the combined properties of OPC and CSA cement in their blends can have practical and valuable implementations, i.e., the encapsulation of radioactive waste, 3D printing, etc. [20,21]. It is believed that in OPC/CSA cement blends, the unique hydration behavior of CSA cement is responsible for the early mechanical properties, while OPC plays a vital role in the later properties. Hitherto, the high content of CSA cement can achieve very high early compressive strength (approximately 30 MPa at 6 h) due to C_4_A_3_$\bar{S}$ being able to rapidly hydrate, as previously reported [22]. Despite the OPC/CSA cement blends having obvious complementary superiority in hydration and microstructure development, the compatibility of OPC and CSA cement is still very controversial. For instance, the hydration of CSA cement is quick due to the ettringite in an alkaline environment, but the deterioration of its later strength was found to have occurred due to the presence of microcracks induced by overexpansion [23,24,25,26]. Therefore, a proper understanding of the shrinkage and expansion behavior of OPC/CSA cement blends is necessary to assure the good control of microcracks and improve the physicochemical properties.

Shrinkage cracks in cement-based materials are a key problem in modern concrete science and technology. The shrinkage caused by expansion can be effectively compensated for by hydrating the CSA cement. However, the effectiveness of CSA cement as an expansive additive must be adequately quantified to determine whether the additional initial cost of each strategy is justified. Recommendations based on findings that were obtained were formulated for the future evaluation of the physicochemical properties of OPC/CSA cement blends. Based on these results, these OPC/CSA cement blends can be used as high-quality repair materials for quickly repairing the engineering of road pavements or old concrete, capable of achieving tremendous economic and environmental benefits due to their outstanding properties.

In this paper, the effect of CSA cement on the compressive strength and volume deformation (i.e., the drying shrinkage and expansion) of OPC mortar was investigated, and the hydration behavior and pore structure of the resulting OPC/CSA cement blends were evaluated. Based on the above results, the expansive effect of CSA cement on the shrinkage deformation of blends was further analyzed, and the microstructure drawbacks were detected through ultrasonic measurements. The findings from this study provided a better understanding of the expansion behavior of CSA cement, achieving an optimal design and controlling the shrinkage behavior of the OPC-based materials.

## 2. Raw Materials and Methods

### 2.1. Raw Materials and Sample Preparation

Ordinary Portland cement (OPC) and calcium sulfoaluminate (CSA) cement with a grade of 42.5 R were used; their physicochemical properties are listed in Table 1. Pure OPC mortar was prepared as a reference mixture. Blended cement mortar was prepared at a 1/3 cement–sand ratio and 0.4 water–binder mass ratio (W/B) by adding 2%, 4%, 6% and 8% CSA cement by mass. River sand with a fineness modulus of 2.2 was used. The mixing proportion of cement mortar is shown in Table 2. Both the cement and water were firstly mixed for 1 min in a mechanical mixer, with the river sand then added to the mix for the other 3 min. The same mixing proportions without sand were used to prepare the blended cement paste for the microstructural analysis. The freshly blended cement mortars were cast into 40 mm × 40 mm × 160 mm molds for the compressive strength test, 25 mm × 25 mm × 285 mm molds for the volume determination and 100 mm × 100 mm × 100 mm molds for the microcrack test, and vibrated using a vibration table to remove air bubbles. The prepared mortars and pastes were cured at 25 °C and a relative humidity (RH) of 95% for 1 day, and were then demolded and stored in water at 25 °C until the testing stages. For the microstructural analysis, only the central parts of the hardened cement paste were collected and then immersed in acetone for 7 days before the test [27,28].

### 2.2. Methods

#### 2.2.1. Setting Time and Compressive Strength

The Vicat apparatus was used to measure the initial and final setting times of the blended cement paste by complying with ASTM C191 [29]. The penetration of the Vicat needle determined the initial and final setting times of the blending cement paste into cement paste at 25 ± 3 mm and 0.5 ± 0.5 mm, respectively. In addition, the compressive strengths of the blended cement mortars were measured at 3 days, 7 days and 28 days using a hydraulic compression machine at a loading speed of 2.5 kN/s. The average value of six replicates was taken as the representative value.

#### 2.2.2. Volume Deformation

The cement mortars with the size of 25 mm × 25 mm × 285 mm were prepared for volume deformation (i.e., the drying shrinkage and expansion), and the length change indicated the volume deformation. All the samples were divided into two parts for the drying shrinkage and expansion. The initial length of the partial samples was recorded after demolding, and then kept in saturated lime water to cure at 22 ± 2 °C for the expansion test, while others were cured in a chamber at 25 °C and an RH of 60% for the drying shrinkage test, respectively. The length changes of the mortar were recorded with a comparator (with an accuracy of 0.001 mm) at regular time intervals, with the average length of three samples taken as the representative values for the drying shrinkage and expansion. The average value of the three tests was taken as the representative value.

#### 2.2.3. XRD

The X-ray diffraction (XRD) analysis (Bryke D8 Advance, Germany) was performed to evaluate the mineralogical composition of the hydration products. The hardened blended paste was vacuum-dried at 20 °C and then crushed and ground until the powder passed through a 45 μm mesh. The accelerating voltage, current, dwelling time and scan increment parameters were 20 kV, 20 mA, 0.2 s and 0.02°, respectively.

#### 2.2.4. Reaction Kinetics and Thermal Analysis

The heat of the hydration for the blended pastes prepared with W/B of 0.4 was monitored using a TAM air isothermal calorimeter at 20 °C for 3 days. The blended cement powder was placed in a glass ampoule and water was introduced into a syringe. After the water was injected, the cement paste was stirred with a mechanical rotor for 2 min in the isothermal calorimeter. The hydration heat was collected using an isothermal calorimeter with an accuracy of 0.001 μW, with the heat flow signals recorded every 60 s for 72 h. In addition, a thermogravimetric analysis and differential scanning calorimetry (TG/DSC, Mettler-1600HT) were used to evaluate the content of the hydration products. Approximately 25 ± 5 mg of the powders were heated under an argon atmosphere from 50 °C to 950 °C, at a heating rate of 10 °C/min.

#### 2.2.5. Pore Structure Characteristics and Microcrack Analysis

The pore structure characteristics of the hardened blends were obtained using mercury intrusion porosimetry (MIP, Quantachrome, AutoPore V9600). During this test, the contact angle and the surface tension of mercury were settled as 130° and 0.48 N/m, respectively. In addition, ultrasonic technology was used to assess the drawbacks of the cement mortar, and the Tektronix AFG3022B signal generator transmitted an ultrasonic signal at a central frequency of 50 kHz with a 70% bandwidth, which was received by a Tekronnix TDS1002B digital oscilloscope. During this test, the generator and receiver were vertically held at the same level on both sides of the cubic samples.

#### 2.2.6. Pore Solution

Cement pastes with a W/B of 0.4 were cast for pore solution extraction, and the de-ionized water was used as the mixing water. These samples underwent the same curing regime as the one used for the prismatic samples. The pore solution was extracted after 3, 7 and 28 days through the accompanying cement under a load of up to 400 MPa using steel method, followed by a centrifugation operation (10,000 rpm). An inductively coupled plasma/optical emission spectroscopy (ICP-OES, Spectro Blue, Kleve, Germany) was used for the evaluation of the Ca, Al and Si ion concentrations. Before the measurement, the solution was digested with a 5% nitric acid solution to prevent carbonation. Based on the measured concentrations of Ca, S and Al, the saturation index of the ettringite was calculated using GEMS to evaluate the driving force of the expansion in the cement blends. The detailed description of the saturation index is given in Ref [30].

#### 2.2.7. SEM/BSE-EDS

A JSM-7610F scanning electron microscope (SEM, JEOL, Tokyo, Japan) was used to observe the morphology of the hydration products. The samples were investigated in a backscattered electron (BSE) mode, and the elemental composition analysis was performed using energy dispersive spectrometry (EDS). The BSE images were observed on polished sections coated with a thin carbon layer for conduction, and the accelerating voltage was set to 20 kV.

## 3. Results and Discussion

### 3.1. Setting Time

Figure 1 shows the effect of the CSA cement on the setting time of the OPC paste. From Figure 1, it could be seen that the setting times of the OPC pastes shortened with the increase in CSA cement. Compared with the pure OPC paste, the initial setting time was shortened by 26%, 45%, 56% and 63%, and the final setting time by 21%, 35%, 46% and 55% when 2%, 4%, 6% and 8% of CSA cement was added, respectively. The setting time of the OPC paste with the addition of 8% CSA cement was shortened by approximately 3 h, which indicated a pronounced accelerated effect of the CSA cement on the hydration of OPC. For the OPC/CSA cement blends, the ye’elimite (C_4_A_3_$\bar{S}$) from the CSA cement hydrated first to form the ettringite [31]. The acceleration effect on the setting time of the blends could be ascribed to the rapid formation of needle-like ettringite, plate-like hydroxyl- or sulfate-AFm and Al(OH)_3_ gels, leading to their rapid stiffening [22,32].

### 3.2. Compressive Strength

Figure 2 shows the compressive strength developments of the OPC and OPC/CSA blended cement mortars at different stages. It can be seen from Figure 2 that the 3 d compressive strength of the blended cement mortar increased with the increase in CSA cement. The highest compressive strength reached 36.8 MPa with an increment of 29.9% when 8% CSA cement was used, indicating the improvement in early strength with the CSA cement. It was attributed to the formation of AFt due to the C_4_A_3_$\bar{S}$ hydration, which enhanced the early strength development. The higher the strength of the AFt cross-linking with C-S-H gels, the easier was it to produce a dense structure and high strength in the blending cement. The 7- and 28-day compressive strengths of the blended cement mortars showed a higher value than that of pure OPC mortar when less than 6% CSA cement was used. Compared to OPC mortar, the compressive strength of OPC mortar incorporated with 6% CSA cement showed an increment in the strength of approximately 41.3% and 30.1% at 7 and 28 days, respectively. However, the excess in CSA cement impacted the 7- and 28-day compressive strengths negatively, and overall reductions of approximately 6.3% and 3.8% were found at higher CSA levels (8%). In addition, no improvement was previously reported in the late compressive strength of the blended cement mortars with a high percentage of CSA cement (more than 25%) [22,26]. Here, the results showed that the rapid hydration of the CSA cement played an essential role in improving the early compressive strength of the blends at low substitution levels (<6%). In contrast, the hydration of OPC was considered to be responsible for the late compressive strength development. The reduction in the late compressive strength might be attributed to the localized expansion that formed microcracks induced by the AFt growth. In addition, due to high reactive C_4_A_3_$\bar{S}$, the prehydration of the CSA cement had a detrimental effect on the hydration and compressive strength gain of the cement blends due to the modification of the surface area and surface charge of minerals, which strongly influenced the hydration activity of the OPC that caused a reduction in the late compressive strength [33,34].

### 3.3. Volume Deformation

The volume deformation characteristics of OPC and OPC/CSA blended mortars as functions of curing time are shown in Figure 3. From Figure 3a, it could be seen that when the samples were cured in saturated lime water, the OPC mortar showed a slight shrinkage deformation up to 7 days, followed by an expansion behavior. The blended mortars showed a continuous expansion with time, and the expansion value increased with the increase in CSA cement. Compared to the OPC mortar, the OPC/CSA cement blends showed expansions in the range of 0.05~0.2% when 2~8% CSA cement was added. Such expansion stress is governed by the hydration of C_4_A_3_$\bar{S}$, which resulted in the formation of the AFt phases. Therefore, it could be deduced that an increase in CSA cement in the OPC/CSA blends increased the AFt formation, resulting in higher crystallization stress. On the other hand, when the samples were cured at an RH of 60% and temperature of 20 °C, the OPC and OPC/CSA cement blended mortars showed an obvious shrinkage behavior (i.e., drying shrinkage). In addition, as shown in Figure 3b, the drying shrinkage of the OPC and blended mortars reached an equilibrium value at approximately 30 days of exposure; however, the total drying shrinkage values decreased with the increase in CSA cement. At 120 days, the total drying shrinkage of the OPC mortar was approximately 350 μm/m. The drying shrinkage of the blended mortars decreased with the increase in CSA cement, and the lowest drying shrinkage value was approximately 250 μm/m when the 6% and 8% CSA cements were used, indicating a beneficial impact of CSA cement in controlling the shrinkage of blends. The reduction of approximately 28% in the OPC with 6% CSA cement was owed to the expansion behavior of the AFt, and the main driving force of the shrinkage reduction was closely related to crystallization stress due to the growth of AFt crystals.

As shown in Figure 3c, the evolution of the drying shrinkage and expansion characteristics of the OPC and OPC/CSA blends could be divided into two periods. Regardless of the curing condition, the OPC mortar exhibited a lower shrinkage deformation before 7 days. The lower shrinkage deformation of the OPC mortar could be attributed to the chemical shrinkage during the hydration process. After 7 days, the expansive deformation of OPC mortar resulted from the osmotic swelling of calcium silicate hydrate (C-S-H) gels. However, before the 40 days of curing, the OPC/CSA cement blends showed a continuous expansion behavior, and the expansion deformation of the OPC was negligible in comparison with those of the OPC/CSA blends. In addition, the expansive effect of the blended cement before 7 days was always higher than those corresponding to 7–40 days. Similar phenomena were also reported, and the OPC/CSA cement blends showed microcracks when the expansion rate was 4~5%, with obvious cracks found when 15% CAS cement was added into the OPC system [31,35]. Figure 3c shows that with an increase in CSA cement, the 0~7-day drying shrinkage of the OPC mortar increased slightly, but the 7~40-day drying shrinkage decreased. The higher drying shrinkage of the blended cement at 0~7 days might have been due to the increase in the chemical/autogenous shrinkage induced by the accelerating effect of CSA cement hydration. It was reported that CSA cement can accelerate the hydration of OPC, and the chemical shrinkage of CSA cement paste was twice that of OPC paste at an early stage [36,37]. The results showed that the drying shrinkage of cement-based materials at an early stage seemed not to be compensated by the crystallization stress of the Aft. However, as shown in Figure 3c, the lower drying shrinkage corresponded to the higher expansion, as shown in Figure 3b, during the 7~40 days, meaning that the 7~40-day drying shrinkage of the OPC mortar could be effectivity compensated due to expansion induced by the CSA cement hydration. The shrinkage compensation of the OPC/CSA cement blends could be explained by the crystallization stress of the AFt crystals, which was closely related to the supersaturation index and capillary pores. It was reported that the OPC/CSA cement blends had a higher supersaturation AFt index compared to OPC, and the supersaturation index decreased with the increase in curing stages [25,31]. Based on the above theory, it could be assumed that at early hydration, crystallization stress from the growth of AFt crystals could have been released entirely due to high porosity, so an insignificant expansion was generated at early hydration. At late hydration, the blended cement showed a lower porosity due to the filling effect of hydration products, so the driving force for expansion was a result of compression due to expansion under the restraint of the pore walls. Results were corroborated by the pore structure characteristics of the blended cement, as shown in TG-DSC results.

### 3.4. Hydration Kinetics

The hydration heat curves of the OPC and OPC/CSA blends are illustrated in Figure 4. Figure 4a shows that there was a larger amount of heat released in the early hydration of the blends than that in OPC paste alone, and the total heat that evolved at early times (approximately 10 h) in blended pastes was 15.7~57.6% higher than that of the OPC paste. However, the total heat release of the OPC paste decreased with the increase in CSA cement, and it was reduced by 13.2~37.5% when 2–8% CSA cement was used. The results illustrated that the early hydration process of the OPC paste was accelerated by incorporating the CSA cement, causing a sharp reduction in setting times, as depicted in Figure 1. From Figure 4b, it could be seen that the initial exothermic peak (peak I) occurred after the addition of water due to the wetting and dissolution of the cement particles, and the initial exothermic peak was closely related to the undersaturation of the solution. During this period, the clinker minerals gradually dissolved into the pore solution, and the pore solution contained a large number of various ions (e.g., Ca, Al, Si and S) with an undersaturation, which would provide the extra energy to accelerate the etch pits on the surface of the particles [38]. After the end of peak I, a long induction period of the OPC paste at approximately 2.5 h was followed, and the rate of dissolution decreased dramatically. The induction period of OPC was shortened by approximately 2 h when more than 4% CSA cement was added. It, thus, could be deduced that the dissolution process of the blends accelerated due to the presence of the highly reactive C_4_A_3_$\bar{S}$. During the induction periods, the ion concentration in the pore solution showed a supersaturation, which would hinder the further dissolution of clinker minerals. Therefore, the induction period could govern the setting time of cement, and the reduction in the setting time resulted from the rapid dissolution of C_4_A_3_$\bar{S}$, which accelerated the formation of the supersaturation of ions in the pore solution. In addition, the exothermic peak II in the OPC paste at approximately 5~10 h corresponded to the hydration of the C_3_S, and the C-S-H and Ca(OH)_2_ began to be produced during this stage. It was noted that the intensity of peak II decreased with the increase in CSA cement, illustrating that C_3_S hydration in the blends was delayed in comparison with the OPC paste alone. The delayed hydration of C_3_S could be supported by the XRD results, as shown in Figure 4. The shoulder peak III that followed peak II in the OPC is usually associated with the conversion of AFt into AFm due to the reduction in sulfate ion concentration [39]. Peak III became more pronounced when 8% CSA cement was used, implying an acceleration in gypsum consumption. The main reason was due to C_4_A_3_$\bar{S}$ from the CSA cement that may have reacted with more gypsum during early hydration, and the absence of sulfate ions resulted in the formation of AFm [40]. Additionally, the sulfate proportion in the blended cement was adjusted to avoid AFm forming, which had been reported in previous studies [22,34,41].

### 3.5. Hydration Products

#### 3.5.1. XRD Analysis

The XRD analysis was performed to qualitatively evaluate the hydration products and the results of the blends, which are shown in Figure 5. From Figure 5, it could be seen that the peak intensity corresponded to the unhydrated minerals (C_2_S/C_3_S) in the blends decreasing with time. At day 3, the peak intensity of AFt increased, but the peak intensity of Ca(OH)_2_ decreased with the addition of CSA cement. The results illustrated that the OPC/CSA cement blends predominantly led to C_4_A_3_$\bar{S}$ hydration, which allowed for more ettringite to form. In addition, the reduction in Ca(OH)_2_ was related to the delayed hydration of C_3_S. Some studies have demonstrated that CSA cement can lead to a delayed effect on the hydration of C_3_S at an early stage [26,42,43], and the pore solution in the OPC/CSA blend showed a lower pH than that of OPC, which could have contributed to the formation of AFt. In addition, the AFm phase as a new phase was found when the 8% CSA cement was used. On day 28, when the 8% CSA cement was added, the peak intensity of the AFt in the blends decreased, which resulted from the transformation of AFt into AFm. The presence of the AFm and AFt phases in the cement was dependent on the availability of sulfate and the alkalinity of the pore solution. The rapid consumption of sulfate was discussed in the section concerning hydration heat. In addition, the Ca(OH)_2_ provided the high alkalinity of the pore solution, and the alkalinity affected the stability of the AFt and AFm. The boundary for the disappearance of ettringite was pH = 10.7 and pH = 11.6 for monosulfate in nonequilibrium conditions [44]. The peak intensity attributed to Ca(OH)_2_ at day 28 showed a decreasing tendency first and later an increase with the increase in CSA cement; the lowest intensity was achieved when 4% CSA cement was added. The presence of Ca(OH)_2_ in the OPC/CSA blended cement with a high percentage of CSA cement increased the pH value of the pore solution, which contributed to the formation of AFm.

#### 3.5.2. TG-DSC

Figure 6 shows the thermal analysis results of the hydration products in OPC and OPC/CSA blended cement at 28 days. The thermogravimetry curves in Figure 6a showed the total weight loss of OPC between 50 °C and 950 °C was 14.13% of the initial total weight, and the OPC/CSA blended cement was 16.12% and 16.6% of the initial total weight. The total weight loss increased with the increase in CSA cement, implying the increase in hydration products of the blends. The major mass loss between 40~250 °C and 400~500 °C was associated with the decomposition of C-S-H/AFt and Ca(OH)_2_, respectively, and during the process of decomposition, endothermic peaks were also found in corresponding DSC curves. Compared with OPC, the content of Ca(OH)_2_ decreased by 25% and 19% when 4% and 8% CSA cements were added, respectively, indicating the lower hydration degree of C_3_S in the blends. A small peak at approximately 150 °C and 680 °C in the blends was likely associated with the AFm and CaCO_3_, respectively. A small endothermic peak at 800 °C was found in the DSC curve and no obvious change was found in the DTG curves, illustrating the recrystallization of the hydration products or unreacted minerals [45]. The decomposition peaks corresponding to the C-S-H gels were enlarged, as shown in Figure 6b. From Figure 6b, it could be seen that the decomposition temperature of the C-S-H gel that formed in the OPC was approximately 110 °C, but the blends showed the highest mass loss peak at approximately 115 °C, implying that the C-S-H gels that formed in the blends were difficult to decompose to some extent. The H_2_O/Si ratio in the C-S-H gels increased essentially linearly with the increasing Ca/Si ratio, which involved the presence of vacant tetrahedral sites, a reduced number of hydroxyl groups and the introduction of a Ca interlayer [39,46]. Overall, more Ca ions were released from the OPC/CSA cement blends, which resulted in the formation of C-S-H gels with a high Ca/Si ratio, as evidenced with the ESD.

#### 3.5.3. Pore Solution Chemistry

Figure 7 shows the concentration of various ionic species in the pore solution at various stages. Figure 7 shows that the Si concentration of the pore solution in the OPC and CAS/OPC blends decreased with the increase in curing stages, but other elemental compositions showed unobvious variations. In addition, the Ca and S concentrations in the pore solution increased with the increase in CSA cement, which was owed to the dissolution of gypsum and C_4_A_3_$\bar{S}$. The driving force for expansion in the OPC/CSA blends was the supersaturation of ettringite, which gave rise to crystallization pressure. To estimate the saturation level of AFt, the ionic concentrations were used to calculate the saturation index, for which the result is shown in Figure 8. As shown in Figure 8a, the pH values of the OPC/CSA cement blends were quite low compared to the OPC. Figure 8b shows the saturation levels of the AFt in the OPC and OPC/CSA blends. It could be seen from Figure 8b that with the extension of the curing time, the saturation index of the OPC/CSA cement blends showed a decreasing trend, illustrating that the expansion characteristics of CSA cement mainly occurred early in the curing period. However, it was noted that the expansion of CSA cement for the shrinkage compensation seemed to be effective in the later curing period, as shown in Figure 3c. The reason was explained in the section about the pore structure. In addition, the OPC/CSA cement showed a higher saturation index compared to pure OPC, and the supersaturation index increased with the amount of CSA cement used. The expansion trend of the cement mortar blends, as shown in Figure 3a,b, complied well with the calculated supersaturation index.

#### 3.5.4. SEM/EDS

Figure 9 shows the SEM images of the OPC and OPC/CSA blended cements after 28 days of hydration. It can be seen from Figure 9 that the hydration products were the amorphous gels and were homogeneously distributed in the OPC paste. For the blended cement, there were some clusters of needle shapes (i.e., AFt crystals) with regular orientations when 4% CSA cement was added, while a large deposit of AFm phases and a connected crack network in the pores were found when 8% CSA was used. The amorphous C-S-H gels in the OPC and OPC/CSA cement blends were intermixed with crystals, i.e., Ca(OH)_2_, AFt or AFm, identified in the XRD results. To further investigate the variations in the composition of these crystals and amorphous gels, the EDS point analysis was carried out with the BSE for the hydration products surrounding the cement grains. The EDS point analysis of all the samples was presented by plotting Si/Ca vs. Al/Ca in Figure 10a. The main hydration products were C-(A)-S-H gels with a Si/Ca ratio of 0.4–0.6 and an Al/Ca ratio of 0.03–0.15, and their mean composition was very similar. The results illustrated that the CSA cement had an unobvious effect on the chemical composition of the C-(A)-S-H gels, and the C-(A)-S-H gel morphology was dominated by its chemical composition, nanostructure and solution environment [47]. It should be noted that the EDS data showed a low Si/Ca ratio, as they contained high amounts of calcium hydroxide. Some data with high Al/Ca and Si/Ca ratios were detected due to the intermixing of Al(OH)_3_ gels and C-S-H gels. In addition, the EDS analyses were plotted for S/Ca vs. Al/Ca, as shown in Figure 10b, and these two lines presented an ideal composition for AFt (Al/Ca = 0.33 and S/Ca = 0.5) and AFm (Al/Ca = 0.3 and S/Ca = 0.25). It was noticed that the Al/Ca ratio never deviated significantly from the theoretical value, so it was unnecessary to postulate the presence of hydroxoaluminate or monocarboaluminate in the interlayer sites [48]. For the OPC and OPC/CSA cement blends, more AFt was scattered near the line and no pure AFt or AFm were detected in the Al/Ca vs. Si/Ca ratio plot, confirming the formation of the AFt/C-S-H or AFm/C-S-H mixture. From the above analysis, the CSA cement was shown to be able to contribute to the formation of AFt, which decreased the drying shrinkage of the OPC; however, the occurrence of the conversion of AFt to AFm was also found when 8% CSA cement was used. According to the EDS data, the average value of the Al/Ca and Si/Ca ratio for C-(A)-S-H was calculated, and the results are shown in Figure 11. The average value of the Al/Ca ratio in the blends showed an unobvious fluctuation with the increase in CSA cement, meaning that a limited Al dissolution formed the C-(A)-S-H gels. However, the average value of the Si/Ca ratio in the blends had the tendency to increase with the increase in CSA cement, illustrating the formation of C-S-H gels with a high content of calcium, as shown in Figure 6.

### 3.6. Pore Structure

The pore size distribution of the hardened OPC and OPC/CSA cement blends is shown in Figure 12. From Figure 12a, it can be seen that the pore size distribution of OPC with the addition of CSA cement was improved, and a significant reduction in the most probable pore size was observed from 57 nm to 42 nm with the increase in CSA cement. However, the pore structure could deteriorate due to overexpansion, and the most probable pore size was 106 nm when 8% CSA cement was used. Figure 12b shows a decreasing trend in the cumulative pore volume when CSA cement was less than 6%, and a high cumulative pore volume was achieved when more than 6% CSA cement was used. More importantly, the amount of pores with a size of less than 0.1 μm showed a small fluctuation with the increase in CSA cement, meaning that the effect of CSA cement on the gel pores was limited. The number of capillary pores with the size of 0.1 μm~10 μm was reduced with the increase in CSA cement, indicating the pore refinement of the blended cement. The ability of pore refinement in the blended cement was owed to the expansive effect of CSA cement decreasing the shrinkage microcracks. According to the crystallization pressure theory, the expansion stress exerted on the pore walls by the impingement of the crystal growth was significant in confined spaces. AFt crystals from the CSA cement hydration grew gradually from the supersaturated solution, and the expansion was caused by the confined growth of ettringite in the available porosity [49,50]. In addition, the growth rate of the crystals was proportional to the reciprocal of the radius of the curved meniscus, and the smaller size micropores with a high curvature could easily contribute to the growth of AFt crystals due to the low free energy of crystallization [30]. Therefore, the slight expansion in the OPC/CSA cement blends was beneficial in decreasing the porosity and improving the compressive strength when less than 6% CSA cement was used. Moreover, the shrinkage behavior of the cement-based materials was related to the stiffness and capillary stress. As expected, the porosity of the cement paste decreased with the extension of the curing age, and more compact materials with higher stiffnesses showed low shrinkage stress. Consequently, the OPC/CSA cement blends with low capillary tension and high stiffness showed a lower drying shrinkage in comparison to the OPC mortar. These reasons could explain the shrinkage-compensating behavior of the CSA cement at later hydration. However, the amount of coarser pores (more than 10 μm) in the blended cement increased when 8% CSA cement was used. The results ascribed the oversized expansion to the destruction of the compact structure of the cement. In this study, the blended cement exhibited a large number of coarser pores when 8% CSA cement was added, but it did not show any visible cracking on the surface of the sample. However, in previous studies [31,51], surface cracks in blended cement easily occurred when a high percentage of CSA cement was used (more than 15 wt.%). Based on the above results, the availability of the nearby capillary porosity provided a safety space that could accept the excess ettringite volume without increasing the local stress [41]. The excessive use of CSA cement would cause overexpansion due to the presence of a large amount of AFt crystals, and was detrimental to the pore structure and the later compressive strength of the cement-based materials.

### 3.7. Ultrasonic Testing

Cement-based materials are usually heterogeneous and multicomponent materials that contain gas–liquid–solid phases, and ultrasonic technology as nondestructive testing could be used to test the microdrawbacks of these cement-based materials, since multiple reflections and diffractions can occur when propagated with microcracks [52,53]. The time domain waveform spectra of the ultrasonic wave in the OPC and OPC/CSA cement blends are shown in Figure 13, showing that there were obvious head wave signals in the time domain waveform, and the significant amplitude ratio of the head and second wave was convenient for judging the propagation velocity of the ultrasonic wave. The transmitting times of the OPC and blended cement-based materials were 100.8 μs, 95.3 μs, 93.2 μs, 84.0 μs and 106.0 μs, while the amplitude values were acquired as 0.404 V, 0.596 V, 1.08 V, 1.37 V and 1.65 V when 0%, 2%, 4% and 6% CSA cements were used, respectively. The shortest transmitting time and the highest amplitude value could be achieved when less than 6% CSA cement was added, illustrating that the incorporation of CSA cement decreased the energy loss of the ultrasonic waveform due to its compact structure. When 8% CSA cement was used, the amplitude of the ultrasonic waves would produce energy attenuation in the propagation process, and, hence, the transmitting time increased with the reduction in frequency intensity, which was ascribed to lead to the formation of more microcrack or drawbacks induced by the oversized expansion.

## 4. Conclusions

For the blended CSA cement, the early performance (e.g., the setting time and compressive strength) was facilitated by C_4_A_3_$\bar{S}$ hydration forming the AFt, while the OPC hydration was the main source for the development of late properties (e.g., the volume deformation). The main conclusions could be summarized as follows:

(1) The CSA cement had a positive effect on the setting time and early compressive strength of OPC due to its ability of pore refinement and the hydration acceleration effect, but the excess of CSA cement impacted the 7- and 28-day compressive strengths negatively, and overall reductions of approximately 6.3% and 3.8% were found at higher CSA levels (8%) due the formation of microcracks induced by expansion.

(2) The drying shrinkage of the OPC mortar decreased with the increase in CSA cement, and a reduction in the total drying shrinkage of approximately 28% was detected when more than 6% CSA was used. The saturation degree of the ettringite was the key factor governing the expansion of the OPC/CSA blends. With the increase in CSA cement, the ettringite saturation that resulted in higher crystallization stress increased, and the compensation behavior for the drying shrinkage occurred during the period between approximately 7 and 40 days.

(3) The pore structure of the OPC/CSA cement had considerable influence on the extent of expansion. The 6% CSA cement refined the pore size distribution of the blended cement due to the slight expansion of AFt, but the incorporation of 8% CSA cement increased the amount of coarser pores due to the formation of more drawbacks or microcracks, resulting from oversized expansion. The drawbacks or microcracks were measured using an ultrasonic pulse velocity test, showing that the 6% CSA cement had the shortest transmitting time and highest amplitude, suggesting a more compact microstructure.

(4) The hydration products of the OPC paste were the mixture of AFt/C-S-H or AFt/Ca(OH)_2_, and the incorporation of CSA cement contributed to the formation of AFt. While the OPC/CSA cement showed a higher Ca/Si ratio in comparison to pure OPC, AFt changed to AFm due to the absence of sulfate when 8% CSA cement was used.

## Figures and Tables

**Figure 1 materials-16-02652-f001:**
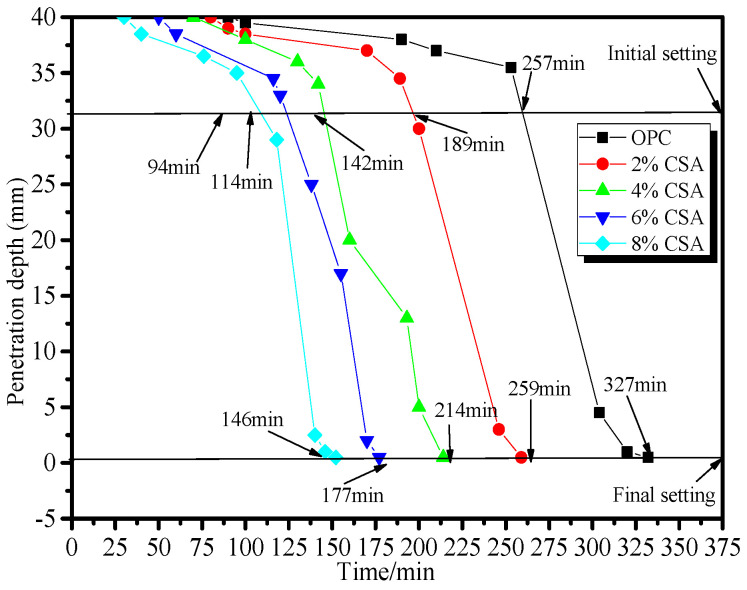
Effect of the CSA cement on the setting time of the OPC paste.

**Figure 2 materials-16-02652-f002:**
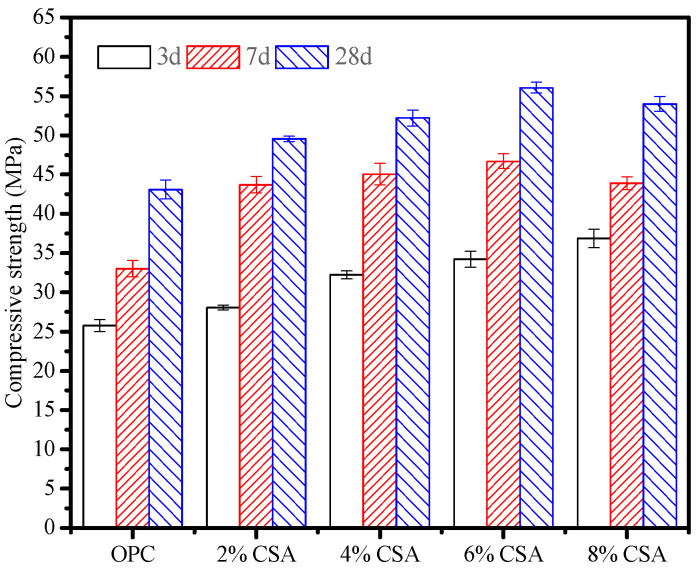
Compressive strength developments in blended cement mortars at different stages.

**Figure 3 materials-16-02652-f003:**
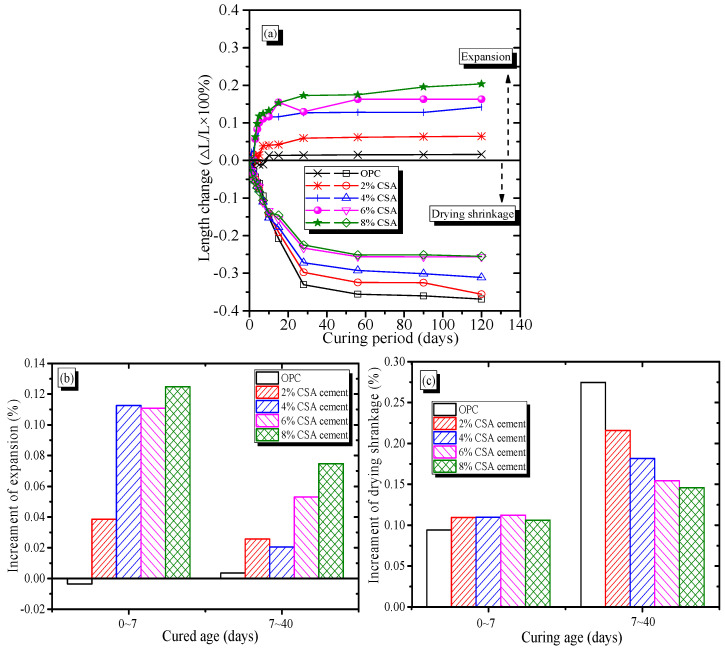
Effect of the CSA cement on the shrinkage–expansion characteristics of OPC mortars, (**a**) The volume change; (**b**) The percentage of expansion; (**c**) The percentage of shrinkage.

**Figure 4 materials-16-02652-f004:**
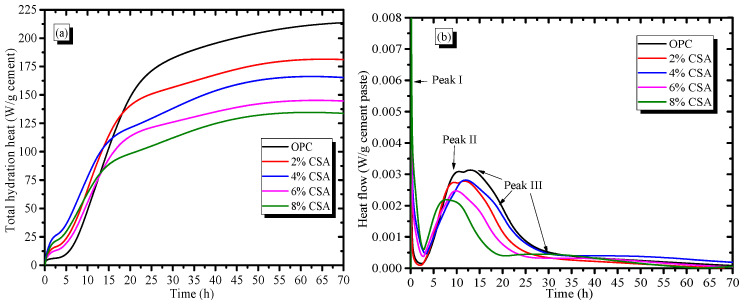
Effect of the CSA cement on the hydration kinetics of OPC, (**a**) The hydration heat; (**b**) The heat flow.

**Figure 5 materials-16-02652-f005:**
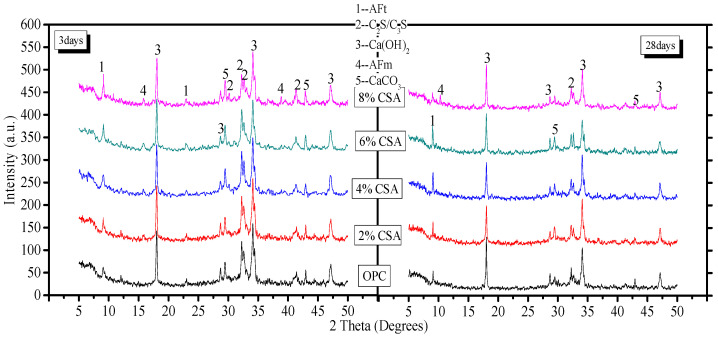
XRD patterns of OPC paste with and without CSA cement.

**Figure 6 materials-16-02652-f006:**
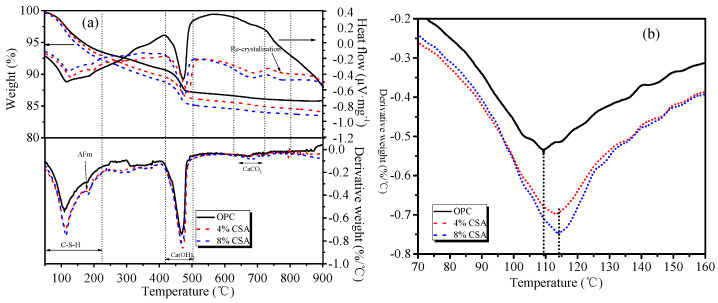
Thermogravimetric analysis of hydration products at 28 days: (**a**) TG-DSC curves and (**b**) the mass loss of C-(A)-S-H.

**Figure 7 materials-16-02652-f007:**
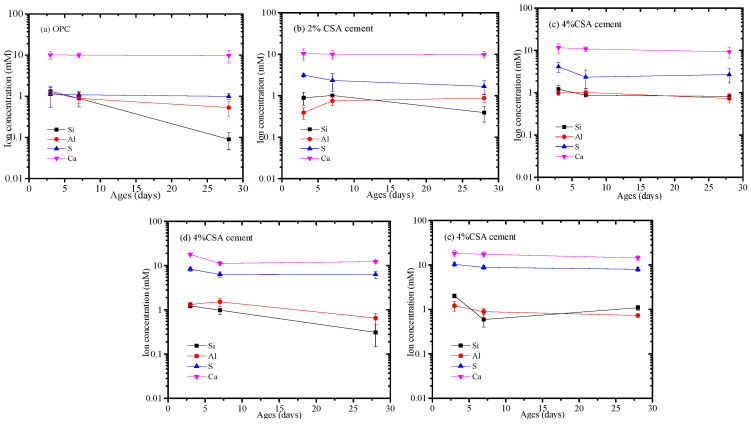
Elemental composition of pore solution for OPC and OPC/CSA blends, (**a**) OPC; (**b**) OPC-2%CSA cement; (**c**) OPC-4%CSA cement; (**d**) OPC-6%CSA cement; (**e**) OPC-8%CSA cement.

**Figure 8 materials-16-02652-f008:**
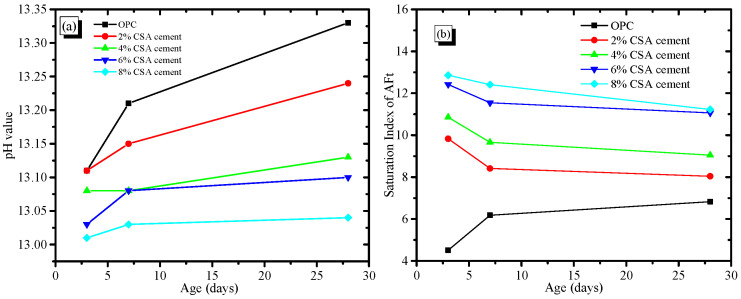
pH value of pore solution and saturation index of AFt in the OPC/CSA blends, (**a**) The pH value; (**b**) The saturation index of AFt.

**Figure 9 materials-16-02652-f009:**
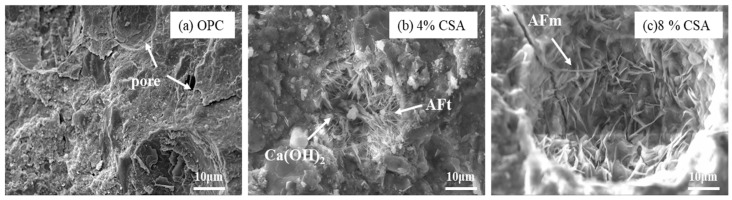
SEM images of OPC and OPC/CSA cement blends at 28 days, (**a**) OPC; (**b**) 4%CSA cement; (**c**) 8%CSA cement.

**Figure 10 materials-16-02652-f010:**
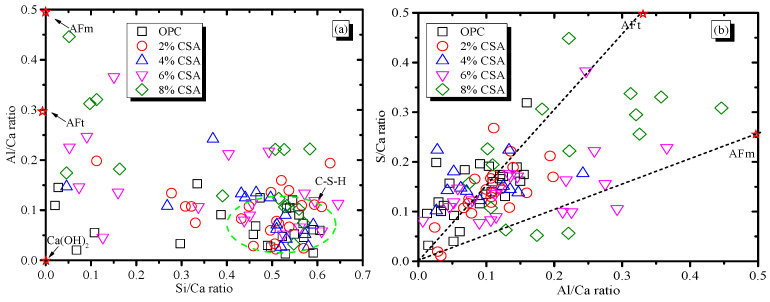
BSE/EDS plots of the atomic ratios for cement, (**a**) Al/Ca vs. Si/Ca (**b**) S/Ca vs. Al/Ca.

**Figure 11 materials-16-02652-f011:**
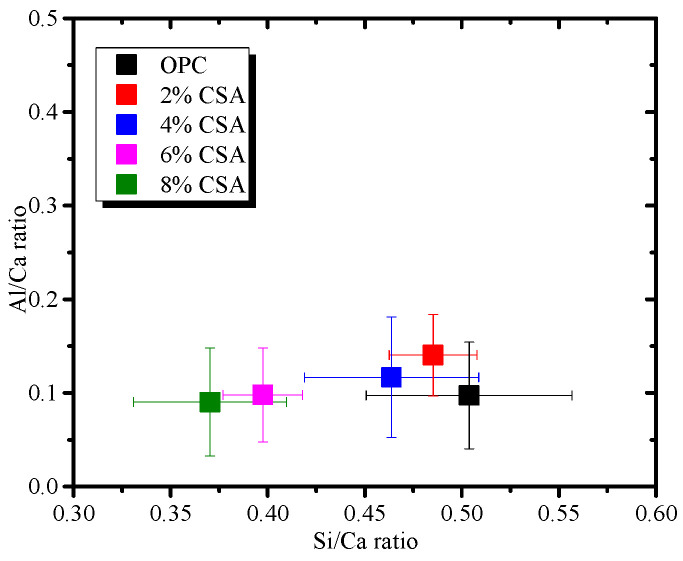
The average value of Al/Ca and Si/Ca in the C-S-H gels.

**Figure 12 materials-16-02652-f012:**
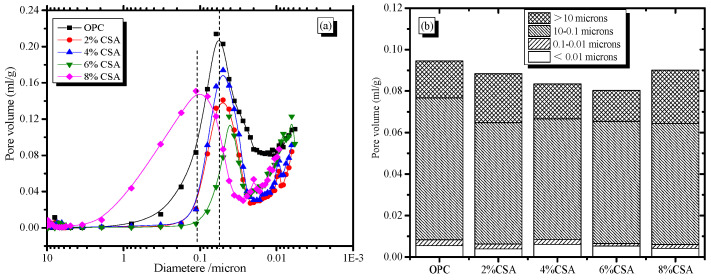
Effect of CSA cement on the pore structure (**a**) Pore size distribution of cement pastes, (**b**) The percentage of pore with different size.

**Figure 13 materials-16-02652-f013:**
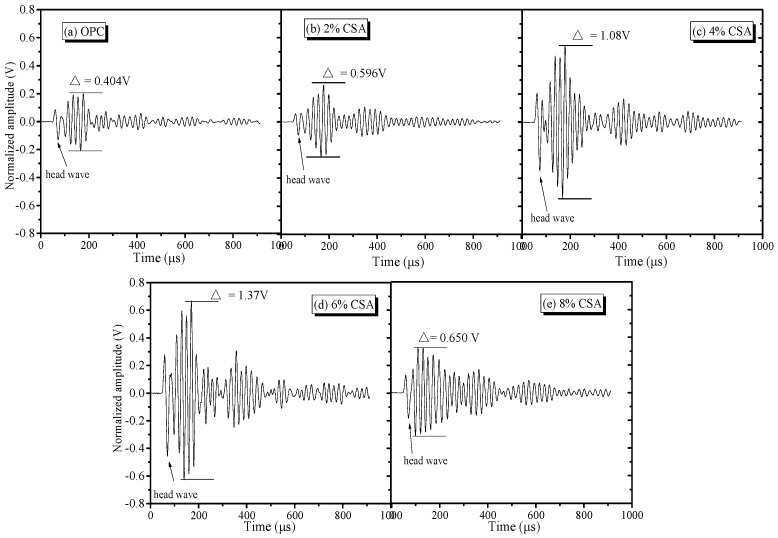
Time domain waveform of the ultrasonic wave of OPC and OPC/CSA cement blends, (**a**) OPC; (**b**) OPC-2%CSA cement; (**c**) OPC-4%CSA cement; (**d**) OPC-6%CSA cement; (**e**) OPC-8%CSA cement.

**Table 1 materials-16-02652-t001:** The physicochemical properties of OPC and CSA cement.

	OPC	CSA
Calcium oxide (CaO)	62.27	44.37
Silicon dioxide (SiO_2_)	20.27	9.16
Aluminum oxide (Al_2_O_3_)	3.77	23.26
Iron oxide (Fe_2_O_3_)	2.99	2.53
Magnesium oxide (MgO)	4.01	4.59
Sulfuric anhydride (SO_3_)	2.50	10.11
Sodium oxide equivalent (Na_2_O)	1.02	0.73
Loss on ignition	0.85	1.16
Fineness (m^2^/kg)	358	313
Normal consistency	320	270
3d compressive strength (MPa)	27.4	37.1
28d compressive strength (MPa)	43.5	44.1

**Table 2 materials-16-02652-t002:** The mixing proportion of cement mortar.

NO.	OPC	CAS Cement	Water/(OPC + CAS Cement) Ratio	(OPC + CAS Cement)/Sand Ratio
1	100%	0%	0.4	1/3
2	98%	2%		
3	96%	4%		
4	94%	6%		
5	92%	8%		

## Data Availability

Not applicable.

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
