# Peer review of "Volume Deformation and Hydration Behavior of Ordinary Portland Cement/Calcium Sulfoaluminate Cement Blends"

_materials, 2023, doi:10.3390/ma16072652_

Round 1

Reviewer 1 Report

Ready to be published.

Author Response

Reviewer 1

Ready to be published.

Response: The authors appreciated the reviewer comments.

Reviewer 2 Report

From a scientific point of view, the paper contains all the necessary information to understand the research results and how they were obtained. The authors have incorporated the suggestions I provided in the first review.

However, from a grammatical and syntactical standpoint, the paper could be significantly improved. There are still some redundant expressions, such as here: Line 375, 377: “In addition, the OPC/CSA cement showed a higher saturation index compared in comparison to the pure OPC, and the supersaturation index increased with the increase of the CSA cement.”

The sentence could be revised as follows: "In addition, the OPC/CSA cement showed a higher saturation index compared to pure OPC, and the supersaturation index increased with the amount of CSA cement used."

Therefore, I recommend analyzing the wording again and, if necessary, reformulating some expressions before publishing it.

Author Response

Reviewer 2

From a scientific point of view, the paper contains all the necessary information to understand the research results and how they were obtained. The authors have incorporated the suggestions I provided in the first review.

However, from a grammatical and syntactical standpoint, the paper could be significantly improved. There are still some redundant expressions, such as here: Line 375, 377: “In addition, the OPC/CSA cement showed a higher saturation index compared in comparison to the pure OPC, and the supersaturation index increased with the increase of the CSA cement.”

The sentence could be revised as follows: "In addition, the OPC/CSA cement showed a higher saturation index compared to pure OPC, and the supersaturation index increased with the amount of CSA cement used."

Therefore, I recommend analyzing the wording again and, if necessary, reformulating some expressions before publishing it.

Response: The authors appreciated the reviewer comments. Please see the attachment, and this sentence has been revised in manuscript as line 383-384.

Reviewer 3 Report

This paper addressed the improved hydration behavior and performance of the CSA/OPC mixture. The expansion behavior of blending cement is evaluated using the saturation index of ettringite. The experimental results are commendable and have been thoroughly carried out. I have a few comments before acceptance:

1. What do the authors mean by induction periods?

2. Can the authors provide the images in the paper of the sample of the CSA/OPC before and after the compression test?

3. The author mentioned that the micro-cracks resulted due to expansion could negatively impact the late compressive strength. In the following line, the authors credited that the expansion behavior of the CSA cement can enhance the resistance against shrinkage cracking. So, what should the readers take from this?

4. In Figure 3, the incremental expansion for OPC is relatively low, which is one advantage of why it is used for building construction with steel reinforcements. However, the expansion is higher for CSA cement compared to its control specimen. How are the authors address this shortcoming? 

5. In Figure 3, the (a) is written as CSA, whereas in (b) and (c) CAS cement is referred. Is this a typo? Similarly, in Figure 4, Figure 6, and Figure 12.

Author Response

Reviewer 3

This paper addressed the improved hydration behavior and performance of the CSA/OPC mixture. The expansion behavior of blending cement is evaluated using the saturation index of ettringite. The experimental results are commendable and have been thoroughly carried out. I have a few comments before acceptance:

  1. What do the authors mean by induction periods?

Response: Thanks for your comments. The induction period is a time of minimal hydration activity between the initial hydration reactions upon wetting and the later primary tricalcium silicate reaction with water to form calcium silicate hydrate and calcium hydroxide. As shown in Fig.4(b), the induction period can be observed between about 1 hours and 2.5 hours, and the rate of dissolution of mineral decreased dramatically. During the induction periods, the ions concentration in the pore solution showed a supersaturation, which would hinder the further dissolution of clinker minerals. Therefore, the induction period could govern the setting time of cement paste. The content was written in Section 3.4. (Please see the attachment.)

  1. Can the authors provide the images in the paper of the sample of the CSA/OPC before and after the compression test?

Response: Thanks for your suggestion. This compressive strength test was carried out by following the ASTM C39. The mortars with size of 40mm×40mm×160mm were prepared. The flexural strength of sample was first evaluated by 3-Point Bend, and then the compressive strength was tested by the compression test. However, based on this method, there is no useful information obtained from the image of samples. Therefore, it is very regret that the image of samples cannot be presented in the manuscripts.

  1. The author mentioned that the micro-cracks resulted due to expansion could negatively impact the late compressive strength. In the following line, the authors credited that the expansion behavior of the CSA cement can enhance the resistance against shrinkage cracking. So, what should the readers take from this?

Response: Thanks for your valuable comments. The drying shrinkage of OPC increased with the increase of CAS cement, and the drying shrinkage of the OPC mortar reduced by 27.8% when 6% CSA cement was used. However, the microcracks due to expansion could impact negatively on the late compressive strength development and associated pore structures of the blends when the replacement content of CSA cement exceeded 6 wt.%. The results relevant to expansion behavior of the CSA cements could induce a crystallization stress, and the less than 6% CSA cement could enhance the resistance against shrinkage cracking. Based on this comment, we have rewritten these contents in the abstract.

  1. In Figure 3, the incremental expansion for OPC is relatively low, which is one advantage of why it is used for building construction with steel reinforcements. However, the expansion is higher for CSA cement compared to its control specimen. How are the authors address this shortcoming? 

Response: Thanks for your valuable comments. As suggested by reviewer, the OPC exposed to the air usually showed a shrinkage behaviour, and the drying shrinkage of OPC mortar would decrease with addition of CSA cement. Therefore, the author thinks that the blending cement is suitable for the building construction with steel reinforcements. However, in Fig.3(a), the OPC and OPC/CSA cement showed expansion behaviour when the samples was kept in saturated lime water, which was a common expansion test method for cement mortar. Based on the reviewer’s suggestion, we had improved the content (as lines 114-117) and it is clearer for reader to understand the advantages of OPC/CSA cement blends.

  1. In Figure 3, the (a) is written as CSA, whereas in (b) and (c) CAS cement is referred. Is this a typo? Similarly, in Figure 4, Figure 6, and Figure 12.

Response: We have modified this Figure.

Reviewer 4 Report

This manuscript has a certain novelty of research and relevance. Quite an interesting research topic that is devoted to the Volume deformation and hydration behavior of ordinary Portland cement/calcium sulfoaluminate cement blends. The paper has been revised, a number of changes and explanations have been made that contribute to improvement, but there are a number of notes:

1) The hardening processes will proceed ambiguously and will differ significantly when using a white-containing cement clinker and an alite-containing cement clinker, therefore, it is necessary to clarify the mineralogical composition of the cement used (or the clinker on the basis of which the cement used was obtained) with an indication of its basicity. In this regard, in the introduction it is necessary to consider the experience of foreign scientists who have been engaged in such research, in particular:

1.1) Kolesnikova O., Vasilyeva N., et all. Optimization of raw mix using technogenic waste to produce cement clinker. MIAB. Mining Inf. Anal. Bull. 2022;(10-1):103—115. [In Russ]. DOI: https://doi.org/10.25018/0236_1493_2022_101_0_103.

1.2) N.N. Zhanikulov, T.M. Khudyakova, B.T. Taimasov, B.K. Sarsenbayev, et.al, “Receiving Portland Cement from Technogenic Raw Materials of South Kazakhstan”, Eurasian Chemico-Technological Journal, vol. 21, no. 4, pp. 333-340, 2019. https://doi.org/https://doi.org/10.18321/ectj890

2) In Figures 1-13, given in the paper, it is necessary to increase the font of the signature of these axes, as well as put down the designations of the figures: a, b, c, etc.

Author Response

Reviewer 4

This manuscript has a certain novelty of research and relevance. Quite an interesting research topic that is devoted to the Volume deformation and hydration behavior of ordinary Portland cement/calcium sulfoaluminate cement blends. The paper has been revised, a number of changes and explanations have been made that contribute to improvement, but there are a number of notes:

1) The hardening processes will proceed ambiguously and will differ significantly when using a white-containing cement clinker and an alite-containing cement clinker, therefore, it is necessary to clarify the mineralogical composition of the cement used (or the clinker on the basis of which the cement used was obtained) with an indication of its basicity. In this regard, in the introduction it is necessary to consider the experience of foreign scientists who have been engaged in such research, in particular:

1.1) Kolesnikova O., Vasilyeva N., et all. Optimization of raw mix using technogenic waste to produce cement clinker. MIAB. Mining Inf. Anal. Bull. 2022;(10-1):103—115. [In Russ]. DOI: https://doi.org/10.25018/0236_1493_2022_101_0_103.

1.2) N.N. Zhanikulov, T.M. Khudyakova, B.T. Taimasov, B.K. Sarsenbayev, et.al, “Receiving Portland Cement from Technogenic Raw Materials of South Kazakhstan”, Eurasian Chemico-Technological Journal, vol. 21, no. 4, pp. 333-340, 2019. https://doi.org/https://doi.org/10.18321/ectj890

Response: Based on the reviewer’s suggestion, we have added the content and cited these REFs.

2) In Figures 1-13, given in the paper, it is necessary to increase the font of the signature of these axes, as well as put down the designations of the figures: a, b, c, etc.

Response: Based on the reviewer’s suggestion, we have modified this Figure (Please see the attachment.).

Round 2

Reviewer 3 Report

The authors have successfully addressed the concerns.